# An Improved YOLOv5-Based Underwater Object-Detection Framework

**DOI:** 10.3390/s23073693

**Published:** 2023-04-03

**Authors:** Jian Zhang, Jinshuai Zhang, Kexin Zhou, Yonghui Zhang, Hongda Chen, Xinyue Yan

**Affiliations:** 1School of Information and Communication Engineering, Hainan University, Haikou 570228, China; whealther@hainanu.edu.cn; 2School of Applied Science and Technology, Hainan University, Haikou 570228, China; jinshuai@hainanu.edu.cn (J.Z.); kysonzhou@hainanu.edu.cn (K.Z.); hdchen@hainanu.edu.cn (H.C.); january@hainanu.edu.cn (X.Y.)

**Keywords:** object detection, YOLOv5, CSPNeXt block, bottleneck transformer

## Abstract

To date, general-purpose object-detection methods have achieved a great deal. However, challenges such as degraded image quality, complex backgrounds, and the detection of marine organisms at different scales arise when identifying underwater organisms. To solve such problems and further improve the accuracy of relevant models, this study proposes a marine biological object-detection architecture based on an improved YOLOv5 framework. First, the backbone framework of Real-Time Models for object Detection (RTMDet) is introduced. The core module, Cross-Stage Partial Layer (CSPLayer), includes a large convolution kernel, which allows the detection network to precisely capture contextual information more comprehensively. Furthermore, a common convolution layer is added to the stem layer, to extract more valuable information from the images efficiently. Then, the BoT3 module with the multi-head self-attention (MHSA) mechanism is added into the neck module of YOLOv5, such that the detection network has a better effect in scenes with dense targets and the detection accuracy is further improved. The introduction of the BoT3 module represents a key innovation of this paper. Finally, union dataset augmentation (UDA) is performed on the training set using the Minimal Color Loss and Locally Adaptive Contrast Enhancement (MLLE) image augmentation method, and the result is used as the input to the improved YOLOv5 framework. Experiments on the underwater datasets URPC2019 and URPC2020 show that the proposed framework not only alleviates the interference of underwater image degradation, but also makes the mAP@0.5 reach 79.8% and 79.4% and improves the mAP@0.5 by 3.8% and 1.1%, respectively, when compared with the original YOLOv8 on URPC2019 and URPC2020, demonstrating that the proposed framework presents superior performance for the high-precision detection of marine organisms.

## 1. Introduction

With the continuous development of computer vision and the exploitation of marine resources, biological detection in underwater environments has entered the public eye, and has been used in underwater robotics [1], underwater detection [2,3], marine research [4,5], and other fields. The autonomous and high-precision detection of marine organisms urgently needs to be realized. However, compared to land-based detection scenes [6,7,8], due to the harsh natural conditions of the seabed when collecting images, most of the images in existing underwater datasets are affected by color casts [9,10,11,12,13], low contrast [14,15,16,17], fuzziness [18,19], noise [20,21], and other quality problems, leading to the loss of clear contour features and partial texture information in the image, which may greatly reduce the usefulness of such images for target recognition. As such, the accuracy of target recognition will be greatly reduced when using such images for target recognition. Increasing the detection accuracy of target marine organisms in complex and changeable underwater environments is still a huge challenge, which has only been considered by a few researchers. Therefore, it is particularly important to propose a high-performance object-detection framework for marine object-detection tasks.

Simple underwater target-detection methods can be better applied when considering underwater conditions such as clear underwater vision, moderate current intensity, and appropriate light. Early traditional detection methods mainly extract features such as color, texture, and geometry. With the continuous development of deep-learning technology, neural networks have been introduced as underwater target-detection frameworks, which can be used to identify and locate objects in images, therefore achieving target detection.

However, the non-ideal underwater conditions in reality result in the degradation of underwater image quality, which, in turn, negatively affects the detection performance. To address the above issues, an underwater object-detection framework based on complex underwater environments can be proposed, which is achieved by embedding the image-enhancement module into the existing detection network framework. First, a series of image-enhancement pre-processing operations are performed on the original underwater image, including CLAHE [22], dynamic threshold [23], multi-scale color adaptive correction [24,25], and other traditional methods, to improve the clarity, contrast, and detailed texture features of the image to improve the image quality, allowing for improved accuracy and better generalization ability of the subsequently trained object-detection model. Then, the powerful object-detection framework based on deep learning can be used to realize the high-precision detection of underwater organisms. In complex and dynamic underwater environments, such as those with low light and poor visibility, the currently popular YOLOv5 detection algorithm still suffers from issues with inaccurate positioning and low detection accuracy for small targets.

As attention mechanisms have continued to develop, many scholars have recently introduced attention mechanisms into object-detection frameworks. A self-attention mechanism [26] was proposed for action classification in a pyramid network; however, the computational burden of the self-attention layer increases quadratically with an increase in image resolution. To overcome this challenge, BoTNet [27] replaces the 3 × 3 convolution in the middle of the Bottleneck used by ResNet50 [28] with MHSA, thus achieving good results in the object-detection task. We note that adding the Transformer self-attention mechanism into a deep network architecture can lead to better detection accuracy, which provided us with the inspiration for improving YOLOv5 in this paper.

Due to the complex and variable nature of underwater environments, the low contrast and obvious color difference in existing datasets make traditional data enhancement approaches inapplicable. According to a large number of studies, image-enhancement operations can improve the quality of images, and the use of different versions derived from image-enhancement operations on original images can increase the diversity of data, thus increasing the quality of a dataset and improving model performance. Therefore, the use of image-restoration methods for data augmentation is a promising method; the authors in [29] also pointed out that adding data-augmentation operations can also improve the detection accuracy of a model.

Research on underwater target detection is of significant importance in the fields of ocean current observation [30], marine biology and security [31]. To enhance the performance of improved YOLOv5, some researchers focus on improving the loss function and detection layer of the backbone module, while relying solely on conventional data-augmentation techniques. In this paper, we propose an improved YOLOv5 object-detection framework to achieve high-precision detection, using the MLLE image-enhancement method [19] to process the original dataset for data enhancement, where the processed images are used as the input to the improved network. The primary contributions of this paper are summarized as follows:The backbone framework of RTMDet [32] is introduced into the YOLOv5 object-detection network. The core module CSPLayer block uses 5 × 5 large convolution kernels, increasing the effective receptive field. The introduced backbone structure improves the performance and robustness of the overall object-detection model.Inspired by BoT in BoTNet, our study introduces a brand-new BoT3 neck network. We introduce the BoT3 module with a self-attention mechanism into the neck network of the YOLOv5 object-detection framework and improve the image feature extraction ability using MHSA, capturing global information and rich contextual information to identify and localize objects accurately.Finally, the improved YOLOv5 detection network architecture was tested on the UPRC2019 and URPC2020 datasets using an MLLE image-enhancement method for data enhancement named union dataset augmentation (UDA). The results of our experiment show that the mAP@0.5 on URPC2019 and URPC2020 after UDA data enhancement reached 79.8% and 79.4%, respectively. The detection accuracy is even improved compared to YOLOv7 and YOLOv8, demonstrating the effectiveness of the proposed network improvement method.

The remainder of this paper is organized as follows: Section 2 discusses the related literature. Section 3 presents the adopted method and network architecture. Section 4 provides the experimental details, while Section 5 presents the experimental results. Section 6 suggests future work. Section 7 summarizes our conclusions and outlook for future research.

## 2. Related Work

### 2.1. Object Detection

Object detection is one of the fundamental and challenging problems in computer vision, with a wide range of applications in various scenarios [33]. Underwater target detection, as one of the most demanding fields in computer science, has broad applications in areas such as aquaculture, autonomous underwater vehicles, and other underwater operations [20] Traditional object detection usually uses statistics-based features; for example, the major idea of the HOG [34] feature detection algorithm is to use the distribution of edge directions to represent the outlines of underwater targets when the edge positions are unknown. DPM [35] is a traditional object-detection algorithm, which artificially designs a convolution kernel, extracts the feature map through various calculations, obtains the excitation effect map, and determines the target position according to the excitation distribution. These methods are intuitive and simple, and can be run quickly; however, their accuracy is low, and the consequent underwater image has obvious color differences and low contrast. Therefore, high-precision real-time detection is not possible with these methods.

As early object-detection methods do not take into account the complexity of vision, there are problems related to poor robustness and generalization ability when dealing with complex environments, limiting their scope of application. Therefore, scholars have gradually introduced deep-learning approaches into object-detection frameworks, allowing for the features at different levels in the input images to be learned by neural networks. In this way, richer image information can be obtained, which is helpful for object recognition by subsequent detection networks. The two-stage detector that first appeared in the public eye is R-CNN [36]. This algorithm first introduced the CNN into the field of object detection; however, its detection accuracy is overly low and the process is cumbersome. Girshick et al. then proposed Faster-RCNN [37], which first extracts the Region Proposal through RPN, then recognizes and predicts the target. Although it has gained better results in most object-detection tasks, its speed still needs to be improved. Next, Mask R-CNN [38] was proposed, extending the original Faster-RCNN by adding the FCN module to achieve the effect of instance segmentation. Overall, R-CNN-based algorithms have an obvious disadvantage: they take a long time and are not suitable when considering complex underwater environments. The YOLO series of algorithms, classified as single-stage detectors, use the convolutional neural network as a regressor and input the entire image to be tested into the neural network. Redmon proposed the YOLO [39] object-detection algorithm, which is a framework with faster training and prediction speed, as well as stronger generalization ability. Subsequently, Redmon proposed the YOLOv2 [40] and YOLOv3 [41] algorithms, which achieved greatly improved detection accuracy based on optimizing the original YOLO. Glenn Jocher et al. proposed YOLOv5 [42], which improves upon the architecture of YOLOv4 [43]. YOLOv5 uses more deep-learning techniques, such as residual network, depth-separable convolution [44], a fine-grained feature pyramid network [45], and so on. Compared with YOLOv4, YOLOv5 not only can achieve better detection accuracy, but it is also faster. However, when working in real complex underwater environments, we still face problems such as light attenuation, more suspended objects, and water flow dynamics. In addition, the detection accuracy will be greatly reduced when a model following such an assumption is used.

To solve the above-mentioned problems existing in underwater environments, scholars have embedded image-enhancement modules into object-detection networks. For example, Zhang et al. [46] proposed a novel object-detection framework that contains an image-enhancement module, following which the detection network is used to perform underwater object recognition and detection. However, the detection network used in this network framework is relatively old, the image-enhancement module network needs to be trained separately, and the required number of calculations is too large. In [47], a combination of the max-RGB method and shades-of-gray method was used to pre-process underwater images. Although the network achieved good performance, it is not suitable for other general datasets. In [48], DG-YOLO was proposed by adding domain-extended datasets of multiple underwater styles to further mine the semantic information of underwater images and significantly improve the detection accuracy. Although the above detection algorithms obtained significantly improved detection accuracy, the accuracy in small-scale object detection remains very low.

### 2.2. A Review of YOLOv5

In this section, we review the network framework of YOLOv5 [42], which provides a basis for its improvement in this paper. YOLOv5 uses an advanced detection framework that allows the network to detect objects at any scale. The basic architecture of YOLOv5 is depicted in Figure 1.

The input module first performs Mosaic data enhancement on the input image, including random scaling and cropping. Immediately afterwards, the adaptive anchor box is computed to determine the best anchor box value for the adaptation. Finally, the input image is adaptively scaled. The backbone network is mainly composed of a focus structure, a CSP structure, and an SPP structure. The focus structure slices the input image, the CSP block is mainly used in the feature extraction part of the CNN to speed up the training and inference process of the model, and the SPP structure strengthens the network’s receptive field. The neck network adopts the structure of FPN and PAN, enhancing the network’s feature fusion ability. The prediction network detects objects in images and predicts their categories and locations.

Since the proposal of YOLOv5, numerous scholars have made various improvements to enhance its model performance. Li et al. [49] proposed GBH-YOLOv5 for PV panel defect detection. The model introduces the BottleneckCSP module, which adds a tiny target prediction head to enhance the detection capability of smaller targets, and employs Ghost convolution to further reduce the inference time. However, this method is not suitable for complex underwater environments. Wen et al. [50] proposed YOLOv5s-CA, which enhances the first C3 module with a greater number of Bottleneck modules and sequentially incorporates attention-based CA and SE modules to refine the YOLOv5s model. Tian et al. [51] improved the YOLOv5 algorithm for high-precision detection of aerial remote-sensing images by integrating CA and BiFPN in the backbone and neck, respectively. Additionally, a small detection head and SIOU loss were incorporated into the model. The improved YOLOv5 achieved high-precision detection of aerial remote-sensing images. Despite the numerous improvements made to YOLOv5 by scholars, these enhanced versions are ill-suited for complex and dynamic underwater environments.

### 2.3. Transformer

Due to the popularity of Transformer, many researchers have introduced attention mechanisms into object-detection networks. Transformer consists of Encoder and Decoder, whose entire network structures are composed entirely of layers with the attention mechanism. Each layer contains sub-modules such as the multi-head self-attention mechanism and the fully connected feed-forward neural network. The attention mechanism is mainly used to improve the performance and speed of the detection algorithm. For example, Yu et al. [52] proposed a novel multi-attention path aggregation network, APAN, which adopts a multi-attention mechanism to further improve the accuracy of multiple underwater object detection. Zhu et al. [53] improved YOLOv5 based on Transformer, replaced the original detection head with Transformer Prediction Heads (TPH), and introduced the CBAM attention module into the proposed network to improve the performance. The BoT used in BoTNet [27] is a computer vision module with a self-attention mechanism, which is mainly used to compress and accelerate the training of neural networks. Its basic principle is the use of a smaller network to simulate the behavior of a large network, to accelerate the training, capture global information, and obtain a good effect in target-intensive scenes. Therefore, in this paper, we introduce an attention mechanism module into the YOLOv5 network framework.

### 2.4. Data Augmentation

Traditional data augmentation is conducted to increase the diversity of images, using methods such as rotation, random cropping, random noise, and traditional image enhancement [54,55,56,57,58,59,60,61,62] without adding additional data. Mosaic enhancement was proposed in YOLOv4 [43]. First, the data of a batch in the dataset is removed, following which four pictures are randomly selected each time to be cut and spliced at random positions to synthesize a new picture. The new data of a batch is obtained by repeating the batch operation many times, thus enriching the background of the objects. TPH-YOLOv5 [53] achieves data augmentation by expanding the datasets of different scenarios, improving the robustness of the model. Similarly, the authors of [48] proposed a WQT dataset extension method that combines multi-domain transformations on the original dataset. Ref. [29] have found that using underwater image restoration as a data-augmentation technique improves object-detection accuracy more effectively than using it solely as a pre-processing step.

## 3. Method

### 3.1. Overall Structure

Here, we propose an improved YOLOv5 object-detection framework. Figure 2 shows the overall framework of the detection network. In this experiment, underwater images are first processed through the underwater image-enhancement module (MLLE framework), following which the image obtained after image enhancement and the original image are used as the input data for the object-detection network. Then, through the improved object-detection framework based on YOLOv5, objects are detected and recognized. Specifically, we introduce the backbone network framework (Section 3.2.1) and improve the Neck module (Section 3.2.2) by introducing the BoT3 module at the medium detection head.

### 3.2. Improved YOLOv5

To improve the detection accuracy, an underwater object-detection method based on the improved YOLOv5 is proposed. For this purpose, we designed a brand-new object-detection framework, introduced the RTMDet backbone network into the backbone, and added the BoT3 module into the neck network. The overall structure of the improved detection framework is shown in Figure 3. Figure 3 illustrates the differences between the classic YOLOv5 and the improved YOLOv5. The blue highlighted area represents the added BoT3 block, while the green dotted frame area depicts the enhanced backbone structure. It was verified experimentally that the improved YOLOv5 framework has better performance and detection accuracy.

#### 3.2.1. Backbone Network

The improved YOLOv5 network uses the backbone framework of RTMDet [32]. The overall framework has a five-layer structure, including one stem layer and four stage layers. The backbone architecture is shown in Figure 4. The stem layer refers to the beginning layer, which includes three Conv blocks. As the number of convolutional layers increases, the model can extract higher-level features and richer information, which helps to improve the downstream dense detection task performance. Stages 1–3 have the same architecture, consisting of one Conv block and one CSPLayer block, while Stage 4 consists of one Conv block, one SPPF block and CSPLayer block.

The CSPLayer block (Figure 5) is the core module of the improved backbone. It consists of three Conv blocks, n CSPNeXt Blocks with residual connections, and one Channel Attention module. The value of n for the CSPNeXt Block in the CSPLayer block used in this article is set to 1. The details of the structure are shown in Figure 6. The green module is the added Channel Attention module, the specific structure is shown in Figure 7. In the improved backbone, it is used to replace the C3 block of the original YOLOv5.

The CSPNeXt Block in CSPLayer introduces a depth-wise convolution with a large 5 × 5 kernel, which expands the receptive field at a reasonable computational cost, and enables the detection network to have a more comprehensive ability to capture context information, which is beneficial for object detection and dense prediction tasks. This indirectly improves the detection accuracy of the model. Furthermore, the large convolution kernel makes up for the quantization error and solves the problems relating to high training cost and quantization difficulty. Figure 6 illustrates the particular structure of the CSPNeXt Block.

When add is true, the Conv block is sequentially connected to the depth-wise convolutional block and added to the input; when add is false, the Conv block is sequentially connected to the depth-wise convolutional block. The depth-wise convolution with a large convolution kernel can significantly increase the effective receptive field to obtain more contextual information, which is critical for object-detection tasks. In addition, large convolution kernels can guide the network to extract more shape features, and target recognition tasks have a higher demand for shape information. Therefore, models that can learn more shape features after introducing large kernel convolutions are more suitable for intensive downstream detection tasks.

The Channel Attention module consists of a AdaptiveAvgPool2d layer, a Conv2d layer, and a Hardsigmoid activation function. The Channel Attention module allows the detection network to focus on meaningful features and weakens the interference of non-key feature information.

In the improved YOLOv5 Backbone framework proposed in this paper, we use the CSPNeXt Block architecture when add is true in the CSPLayer block, and the number n of CSPNeXt Block blocks is set to 1. As off-the-shelf deep-learning tools (e.g., Pytorch) tend to have poor support for large-scale DW convolution [63], to increase the receptive field while ensuring the efficiency of convolution with respect to the GPU, we used a 5 × 5 kernel in CSPNeXt Block normal convolution. If one can find a way to optimize the CUDA kernel to improve computational efficiency, it is recommended to use DW convolution.

#### 3.2.2. Neck Network with Bottleneck Transformer

The improved YOLOv5 neck network is shown in Figure 3. In this paper, the BoT3 module is added to the medium detect head in the YOLOv5 neck framework, which does not change the size of the eigenvector. BoT3 is the core module for improving YOLOv5, consisting of three Conv blocks and a BoT (Bottleneck Transformer); its specific structure is shown in Figure 8. C3 denotes a CSP module that includes three Conv blocks. BoT3 is borrowed from the C3 structure (Figure 3), replacing two CBS convolution blocks in C3 with a BoT module (which is named BoT3).

The concrete structure of BoT is presented in Figure 9. The extraction part denotes the feature extraction layer, implemented by the Conv block, and the expansion part denotes the reshape layer, which redefines the shape of the matrix through the function view. The BoT block uses the self-attention mechanism to improve the model performance, replacing the 3 × 3 spatial convolution with multi-head self-attention (MHSA) in the ResNet Bottleneck module, which extracts feature information through the global attention aggregation convolution block. The BoT block has the advantages of both the CNN and the self-attention mechanism, which allows it to focus on more meaningful content and locations, establishing global dependencies while extracting local information through convolution. Compared to the ResNet Bottleneck, BoT blocks do not employ excessive convolution operations, thus greatly reducing the number of parameters and minimizing latency overhead.

MHSA (Figure 10) adopts a multi-head self-attention mechanism, with input size of *H × W × d*, with *H*, *W*, and *d* representing the height and width of the input feature matrix and the dimension of a single token, respectively. The input data are subjected to 1 × 1 convolution operations to obtain query, key, and value encodings, following which ${\mathrm{qk}}^{T}$ is obtained by matrix multiplication operations for query encoding and key coding. Then, after the SoftMax operation, the result is multiplied by a matrix with value coding to obtain the output. Positional encodings and the value projections in the highlighted blue areas are not present in the non-local layer [64,65]; that is, there is no position embedding in the non-local layer. Therefore, the framework used in our experiment does not take into account position embedding. The MHSA used in this experiment focuses on content information and does not consider content positions; that is, the query code obtained by $W_{Q}$: 1 × 1 is not multiplied by the relative position code to obtain ${\mathrm{qr}}^{T}$.

With the introduction of the BoT3 block, the improved YOLOv5 framework can learn and capture context information more effectively, as well as accurately locate and identify targets by extracting more relevant information, thus boosting the accuracy of model object detection. As a result, the improved detection framework can detect more complex and dense targets.

### 3.3. Dataset Augmentation

We use the MLLE method for the enhancement of underwater images. The MLLE method is exclusively used during the data-augmentation stage and is not applied during the final testing. The processing of underwater images involves the reduction of blue–green tones, improving contrast, and maintaining detail, which leads to a good visual effect. This method consists of two main steps: (1) Local adaptive color correction and (2) local adaptive contrast improvement.

Figure 11 displays images before and after processing by MLLE. The processed images can be seen to have reduced blue–green tones, improved contrast, and reduced noise, thus improving visibility and maintaining good texture with better image quality.

The dataset used in this paper was obtained by the joint expansion of the dataset processed by the MLLE image-enhancement method and the original dataset using a data-augmentation method named UDA. The augmented dataset was used as input to the object-detection network, which improved the diversity of the input data and greatly enhanced the detection accuracy.

## 4. Experimental Details

### 4.1. Datasets

#### 4.1.1. URPC2019

We used the public Underwater Robot Picking Contest 2019 (URPC2019) dataset to evaluate the effectiveness of our proposed marine biological object-detection framework. The URPC2019 dataset includes 3765 training samples and 942 validation samples covering five water target categories: Echinus, starfish, holothurian, scallop, and waterweeds. Examples of the images in the dataset are shown in Figure 12. In this paper, the image size is changed to 416 × 416 pixels by compression operations. In this experiment, 942 raw underwater images participating in the verification were used as a test set, with which we evaluated the performance of the overall framework.

#### 4.1.2. URPC2020

We also used the publicly available URPC2020 [66] dataset, presented for the 2020 Underwater Robot Competition Underwater object-detection Algorithm Optics Competition. There are 4200 randomly selected images in the training set of this dataset, 800 images in the validation set, and 1200 images in the test set, including echinus, starfish, holothurian, and scallop categories and object images featuring various underwater scenarios. Examples of the images in the dataset are shown in Figure 13. In this paper, the image size is changed to 416 × 416 pixels by compression operations.

### 4.2. Evaluation Indication

To comprehensively and objectively evaluate the performance of the proposed model, the mean average accuracy (mAP) was used to measure the accuracy of the model, and the object-detection results were evaluated. Precision represents the ratio of all boxes that are correctly predicted to all boxes that are ultimately output by all networks, defined as follows:(1)$$\mathrm{Precision}=\frac{\mathrm{TP}}{\mathrm{TP}+\mathrm{FP}},$$
where *TP* is the number of positive samples that are correctly identified and *FP* is the number of negative samples that are incorrectly identified as positive samples.

Recall represents the ratio of all predicted correct boxes to all true boxes, defined as follows:(2)$$\mathrm{Recall}=\frac{\mathrm{TP}}{\mathrm{TP}+\mathrm{FN}},$$
where *FN* is the number of positive samples that are improperly identified as negative samples.

AP represents the average accuracy for a certain category with an IoU threshold between 0.5 and 0.95. It corresponds to the area under the PR curve, which is an indicator for a single category:(3)$$\mathrm{AP}=\int_{0}^{1}\mathrm{PRdr}.$$

mAP represents the average AP with respect to all categories, defined as follows:(4)$$\mathrm{mAP}=\frac{1}{C}\sum_{i=1}^{C}{\mathrm{AP}}_{i}.$$
where *C* represents the number of categories in the dataset. The higher the mAP value, the better the model performance.

### 4.3. Experiment Settings

The experimental environment for this study involved training the model on a Tesla V100 SXM2 32 GB GPU, using the GPU driver for Ubuntu 18.04. The model was implemented in Python 3.6 and used CUDA version 10.2 and cuDNN version 8.1. All experiments described in the article underwent 100 epochs of training.

The YOLOv5x model from the YOLOv5 series was used for training, with a batch size of 64 for 100 epochs. The model was trained using a learning rate of 0.01, SGD momentum of 0.937, optimizer weight decay of 0.0005, and initial weights path of yolov5s.pt. All other training parameters were set to the default values of the YOLOv5 network. In our comparative experiment, the YOLOv7 training parameters were set to a model size of yolov7. Limited by the storage capacity of the GPU, for YOLOv8, we used a batch size of 8, a model type of YOLOv8x, and a pre-training model of yolov8x.pt. For the Faster-RCNN model, we employed a pre-trained ResNet-101 model. Similar to YOLOv5, all other parameters for these networks were set to their default values.

## 5. Experiment

To quantitatively evaluate the performance of the proposed overall framework, the proposed object-detection framework was tested on two underwater datasets: URPC2019 and URPC2020. The importance of the CSPLayer and BoT3 modules in the model was evaluated through an ablation experiment, in order to understand their effect on model performance. We also compared the proposed model with state-of-the-art object-detection frameworks, which allowed us to prove that the object-detection framework proposed in this paper has higher accuracy than other popular object-detection frameworks.

### 5.1. Experimental Results Obtained with Improved YOLOv5 on URPC2019

In this paper, the model was trained for 100 epochs and the training results of the improved model on URPC2019 were obtained. The different performance indicators on the training set and validation set are provided in Figure 14 below.

The first three columns show the box loss, object loss, and classification loss of the improved YOLOv5 model, which indicates how well an algorithm predicts the quality of an object. The first three columns display the three loss curves, with their x-axis representing the epochs of the training set and the y-axis representing the overall loss value. From the curves, it is evident that the overall loss value consistently decreases as the training progresses, eventually stabilizing. These results indicate that the improved YOLOv5 model proposed in this paper exhibits a good fitting effect, high stability, and accuracy. The last two columns are PR curves, with their *x*-axis representing the training epochs and the *y*-axis representing precision and recall, respectively. showing the evaluation of object-detection performance when changing the confidence level threshold: the closer the curve value is to 1, the higher the confidence level of the model. It can be seen, from Figure 14, that it was valid to propose the improved YOLOv5 model.

The confusion matrix of the proposed model is shown in Figure 15, which describes the prediction accuracy of our improved YOLOv5x model on the five types of underwater organisms in the dataset, as well as the relationships between the predictions. In the figure, the row direction represents the true label, the column direction represents the prediction category, and the diagonal data represent the correct detection rates. It can be seen that our model obtained a high accuracy rate for each category.

The PR curves of the proposed model are presented in Figure 16. It can be intuitively seen that the rate of change in accuracy increases as the recall increases. As can be seen from Figure 16, the PR curves for the proposed model are all close to the upper right corner, indicating that the recall and accuracy of the proposed framework are high. The large area under the PR curves indicates that our model performs well. Moreover, the PR curves are smooth, indicating that the relationship between the recall rate and accuracy of our proposed model is relatively stable.

### 5.2. Ablation Experiment

In this section, we conducted ablation experiments to verify the effectiveness and reliability of the added method in improving YOLOv5, and evaluated the impact of these improved methods on the experimental results by selectively removing them. Table 1 provides the results of the ablation experiment on URPC2019. “✔” indicates that the improvement method is used. The curve in Table 1 is shown in Figure 17. In Table 1, the detection results of the original YOLOv5 network are shown in the first row of the table, which were used as the baseline in the ablation experiment. The results of the improvement to the original YOLOv5 when using the CSPLayer module (which introduces large convolution kernels) are shown in the second row of the table, the introduction of this module enables the detection network to extract deeper semantic features, resulting in a 0.7% and 1.3% improvement in mAP@0.5 and mAP@0.5:0.95 values, respectively, than those of the original YOLOv5 without any improvement strategy. Then, the results when only adding the BoT3 to improve the accuracy results of YOLOv5 prediction and classification of targets are shown in the third line of the table, since BoT3 has a self-attention mechanism, the improved network has a more comprehensive ability to obtain context information, which leads to the mAP@0.5 and mAP@0.5:0.95 of the model were improved by 1.2% and 0.7%, respectively. Next, the results when only using UDA for data augmentation for object detection are shown in the fourth row of the table, due to the accuracy of object detection in underwater images can be improved through data-augmentation techniques applied to image restoration, the mAP@0.5 and mAP@0.5:0.95 of the model were improved by 1.3% and 4.2%, respectively, compared with the baseline. The fifth line of the table shows the experimental results of YOLOv5 using both the CSPLayer and BoT3 modules, where the mAP@0.5 and mAP@0.5:0.95 were improved by 1.6% and 1.4%, respectively, compared with the baseline. The final detection results of our object-detection framework are shown in the sixth line of the table, where the mAP@0.5 and mAP@0.5:0.95 increased by 4.5% and 2.9% from the baseline, respectively. These data demonstrate that our improvements made to YOLOv5 were effective in improving detection accuracy.

### 5.3. Comparative Experiment

In this paper, we present a comparative analysis of our proposed approach with current popular and state-of-the-art (SOTA) methods. Our evaluation is conducted on the URPC2019 dataset, with the original YOLOv5_x model as the baseline. The results of the comparative experiment on the URPC2019 dataset are shown in Table 2 and Figure 18. Our improved YOLOv5 detection framework achieves mAP@0.5 of 79.8%, which is significantly better than all original YOLOv5 models. Furthermore, it outperforms YOLOv7 and YOLOv8 by 2.3% and 3.6%, respectively, as well as Faster-RCNN by 2.72%. Based on these results, our proposed novel YOLOv5 network architecture demonstrates superior performance.

Moreover, we conducted comparative experiments on the URPC2020 dataset, and the experimental results are shown in Table 3 and Figure 19. The mAP@0.5 of the improved YOLOv5 detection framework proposed in this paper reached 79.4%, which was 1.4% higher than YOLOv5 (baseline), 0.1% higher than YOLOv7, 14.1% higher than Faster-RCNN, 10% higher than the model in another previous study [46], and 1.1% higher than YOLOv8. Similarly, the data demonstrated that our proposed YOLOv5 object-detection framework had higher accuracy than general object-detection methods, proving the effectiveness of our improvements.

The parameters and flops of our improved YOLOv5x are 208.0 M and 432.8 G, respectively. The original YOLOv5x had parameters and flops of 86.7 M and 205.7 G, while YOLOv8x had parameters and flops of 68.2 M and 257.8 G. Despite having a larger number of parameters than the others, our model achieved a higher accuracy than theirs.

To verify the feasibility of the improved model, multiple images from the test dataset were selected. Figure 20 shows the detection results obtained by the improved and original YOLOv5 algorithms in different scenarios. In Figure 20, the first column shows the Ground Truth image, the second column shows the results of the original YOLOv5 detection, and the third column shows the detection results of the improved YOLOv5 proposed in this paper.

Compared with our improved YOLOv5 algorithm, the original YOLOv5 had missed and false detections. There was an occluded detection target in the first row of images, and the improved YOLOv5 algorithm in column c correctly detected the scallop that was not detected by the original YOLOv5 algorithm in column b. The second and fifth rows show the detection of small targets. It can be seen that the improved YOLOv5 detected the small-sized target scallop missed by the original YOLOv5. The third, fourth, and seventh rows show the object-detection performance in underwater blurred and low-visibility scenes. As can be seen from the display diagram, the improved YOLOv5 network detected the echinus and scallop missed by the original YOLOV5. The sixth row shows the object-detection performance for a scene where the targets are dense and there is background interference. It can be seen, from column b, that the original YOLOv5 algorithm had missing and false detections of scallops, while column c shows that the detection results obtained by the improved YOLOv5 algorithm were consistent with the Ground Truth, which could better perform cross-object detection with higher accuracy under complex environments.

In summary, for dense, mutually occluded, and small-scale targets, the improved YOLOv5 network can reduce the missed detection rate and improve the false detection rate, allowing for effective detection. In addition, the improved YOLOv5 network still maintained good detection performance in the case of low underwater lighting contrast and complex backgrounds. From the experimental results, the improved YOLOv5 object-detection algorithm performs better than state-of-the-art models, improves the accuracy of object detection, basically meets the needs of underwater object-detection tasks, and has practical application value.

## 6. Future Work

In future work, there is potential to further investigate improved image-enhancement methods for data augmentation, and explore more suitable transformer blocks to enhance the network structure and improve detection accuracy. The underwater environment is complex and changeable, and conventional object-detection algorithms still have the problem of missing and wrong detection under the conditions of dense targets, complex backgrounds, a wide variety of marine organisms, and the mimicry of similar species. For the improved YOLOv5 proposed in this paper, we still have the problem of the number of network parameters in the proposed object detection being still high, leading to a very large computational cost. Therefore, in the future, we intend to further investigate network compression to reduce network parameters and improve detection speed while ensuring detection accuracy.

## 7. Conclusions

Considering the complex and changeable underwater environment, we propose a new and improved YOLOv5 underwater biological detection framework in this paper. First, the RTMDet backbone framework was introduced, which allows the detection network to extract higher-level features and reduce the interference of non-key feature information. Second, by adding the BoT3 module to the object-detection head in the Neck module of YOLOv5, the accuracy of underwater object detection was improved to identify and locate targets more accurately. Finally, the data of the training set were enhanced using the MLLE image-enhancement method. Through experiments, we found that the mAP@0.5 and mAP@0.5:0.95 of the framework on the URPC2019 dataset reached 79.8% and 44.2% and were 4.5% and 2.9% higher than those of the original YOLOv5, while the mAP@0.5 of the framework on the URPC2020 dataset reached 79.4% and was 1.4% higher than that of the original YOLOv5. In summary, the proposed improved method has significantly better object-detection accuracy than YOLOv5, making it a highly practical general underwater object-detection framework. The improved YOLOv5 framework has shown good performance in dense object detection in underwater environments and can be widely applied to marine object detection in complex underwater environments.

## Figures and Tables

**Figure 1 sensors-23-03693-f001:**
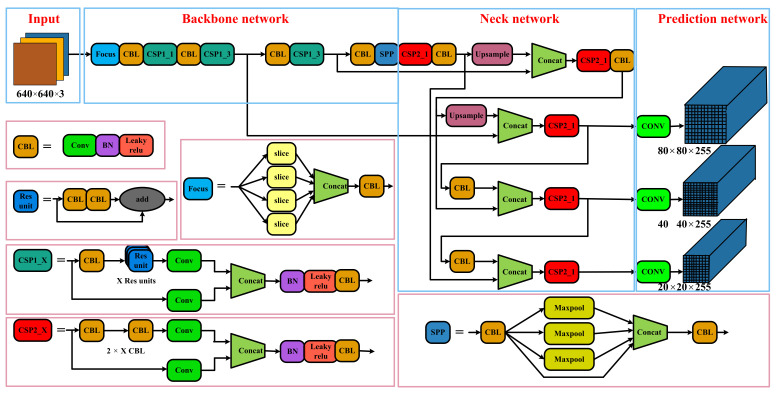
The basic framework of YOLOv5, consisting of four modules: The input module, backbone network, neck network, and prediction network.

**Figure 2 sensors-23-03693-f002:**
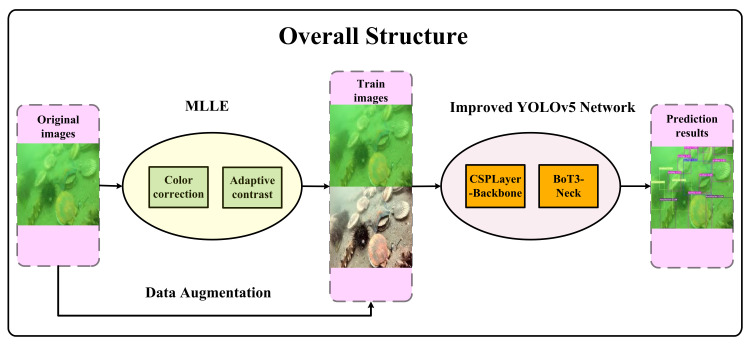
The general object-detection framework.

**Figure 3 sensors-23-03693-f003:**
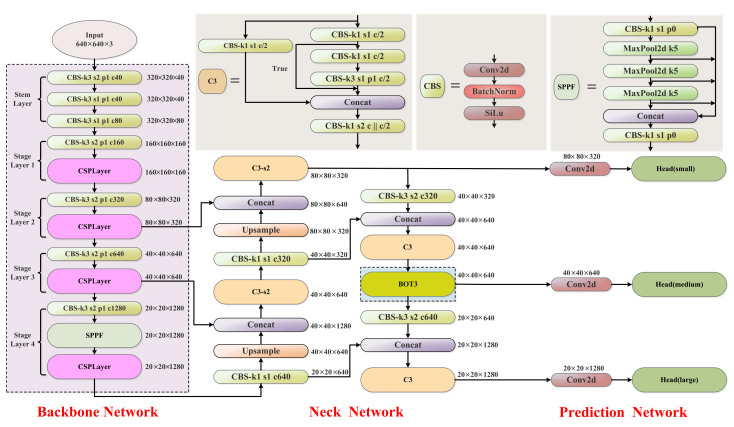
Improved YOLOv5 network framework.

**Figure 4 sensors-23-03693-f004:**
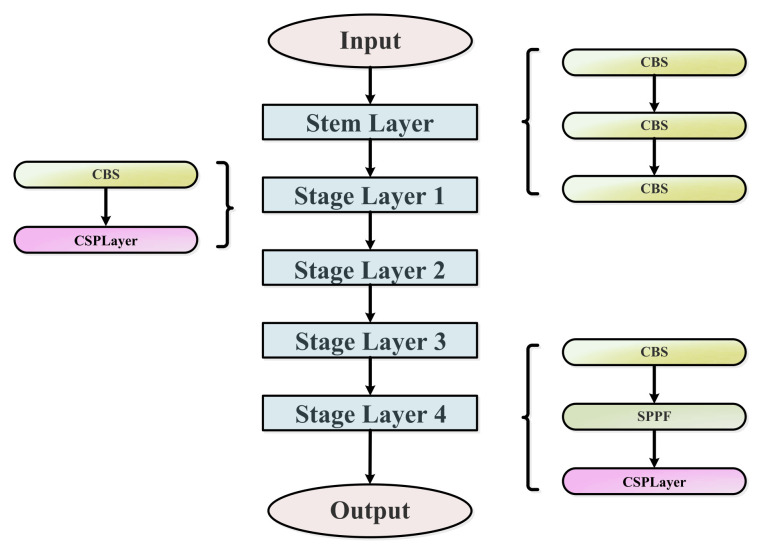
The overall structure of the backbone.

**Figure 5 sensors-23-03693-f005:**
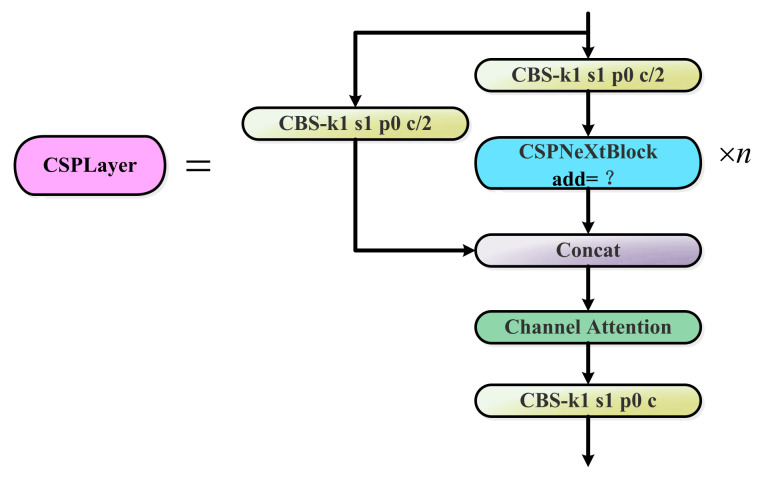
CSPLayer block structure. The blue module is the CSPNeXt Block is improved by introducing a large convolution kernel.

**Figure 6 sensors-23-03693-f006:**
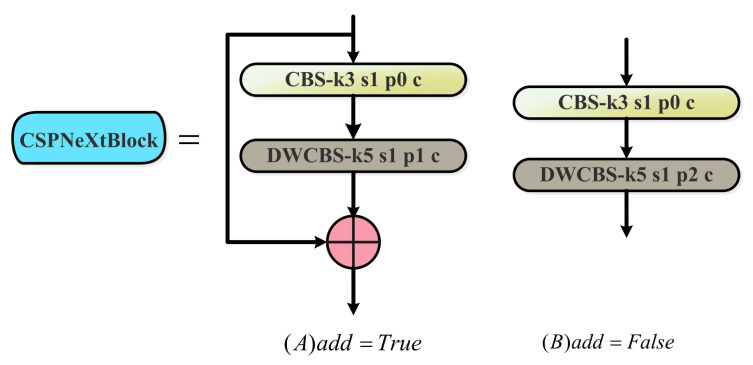
The CSPNeXt Block framework is divided into two different structures, A and B, according to whether add is True or False.

**Figure 7 sensors-23-03693-f007:**
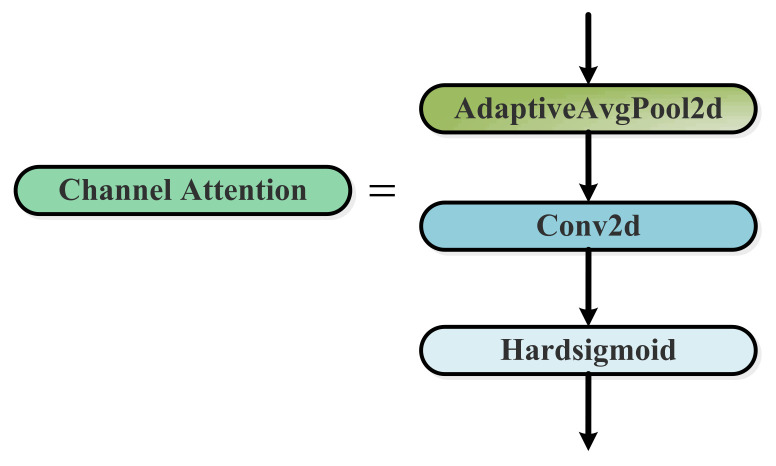
The framework of the Channel Attention block.

**Figure 8 sensors-23-03693-f008:**
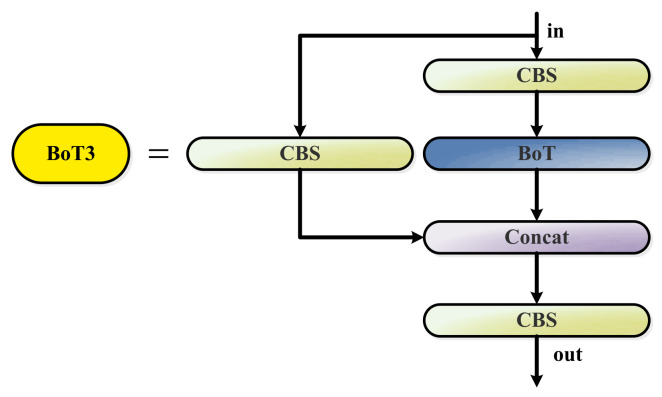
BoT3 added to the Neck of YOLOv5. BoT3 replaces two CBS blocks of the C3 block with the BoT block (Figure 9).

**Figure 9 sensors-23-03693-f009:**
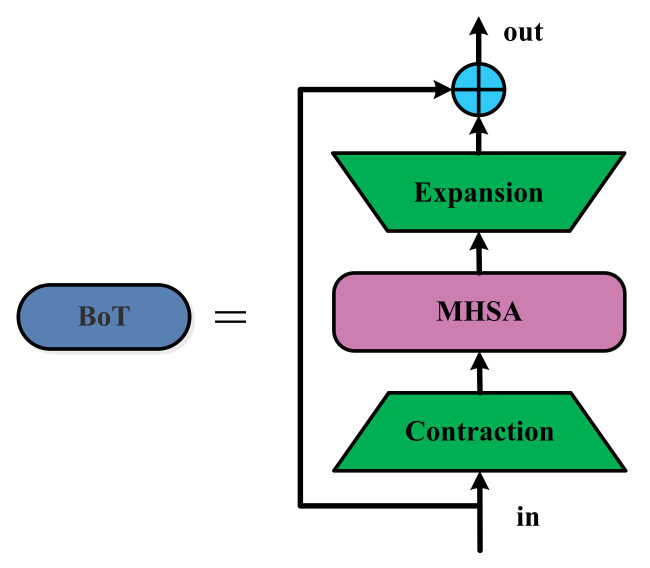
The BoT block in BoT3. The detailed structure of the MHSA is shown in Figure 10. ⊕ stands for the element-wise sum operation.

**Figure 10 sensors-23-03693-f010:**
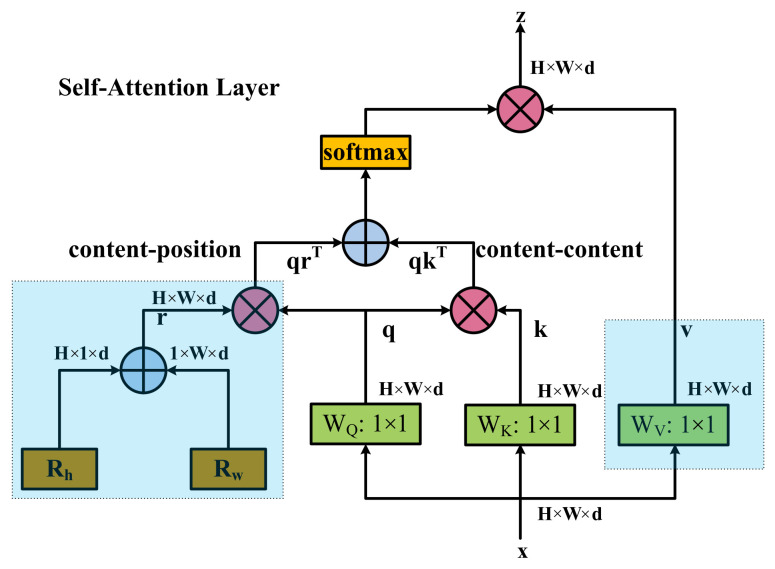
MHSA applied to BoT3. MHSA also uses four heads. q, k, and v represent query, key, and value encodings, respectively. ⊕ and ⊗ represent element-wise sum and matrix multiplication operations. 1 × 1 represents 1 × 1 convolution.

**Figure 11 sensors-23-03693-f011:**
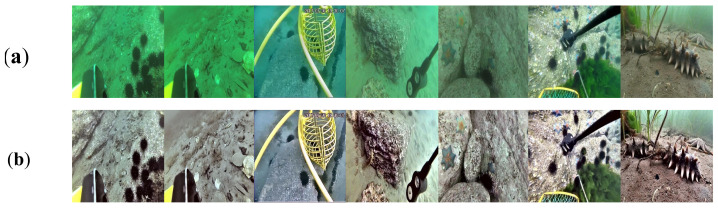
The underwater raw images and images produced by MLLE image enhancement. (**a**) shows the underwater images without any processing, while (**b**) shows the underwater images after image enhancement.

**Figure 12 sensors-23-03693-f012:**
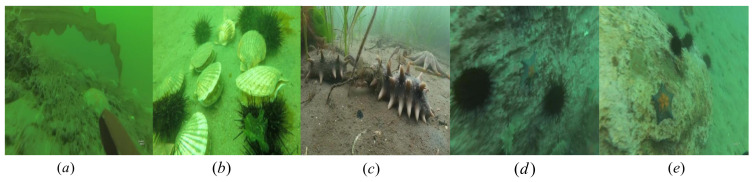
URPC2019 dataset includes five biological categories, namely (**a**) waterweeds, (**b**) scallop, (**c**) holothurian, (**d**) echinus and (**e**) starfish.

**Figure 13 sensors-23-03693-f013:**
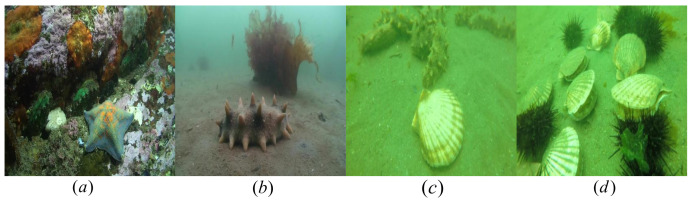
URPC2020 dataset includes four biological categories, namely (**a**) starfish, (**b**) holothurian, (**c**) scallop, and (**d**) echinus.

**Figure 14 sensors-23-03693-f014:**
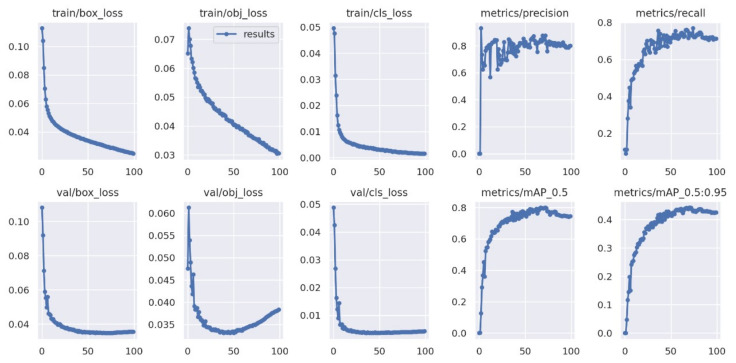
Performance values for improved YOLOv5 model.

**Figure 15 sensors-23-03693-f015:**
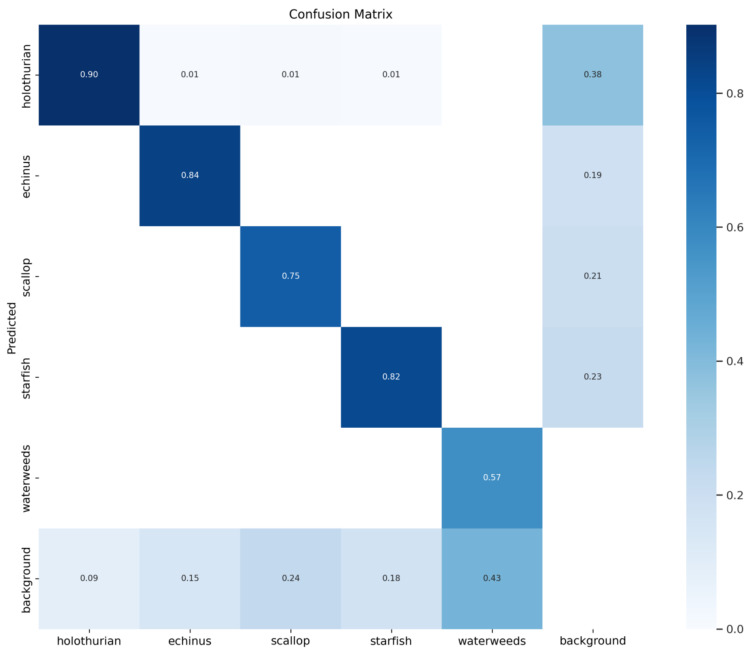
Confusion matrix for the proposed model.

**Figure 16 sensors-23-03693-f016:**
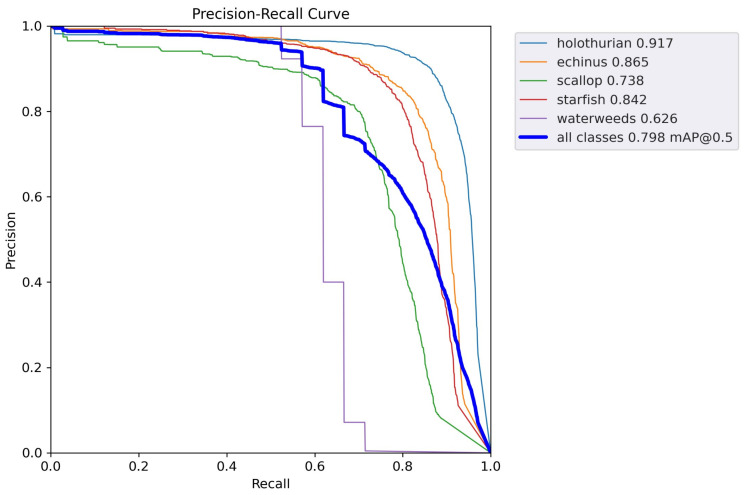
PR curves for the proposed model.

**Figure 17 sensors-23-03693-f017:**
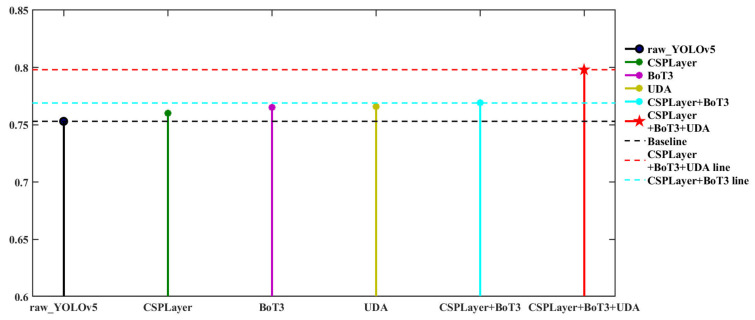
Visualization of results of ablation experiments on URPC 2019.

**Figure 18 sensors-23-03693-f018:**
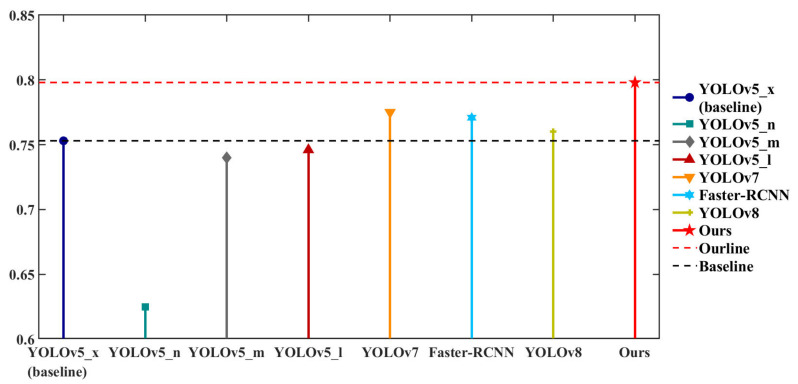
Visualization of comparative experiments based on URPC2019 dataset.

**Figure 19 sensors-23-03693-f019:**
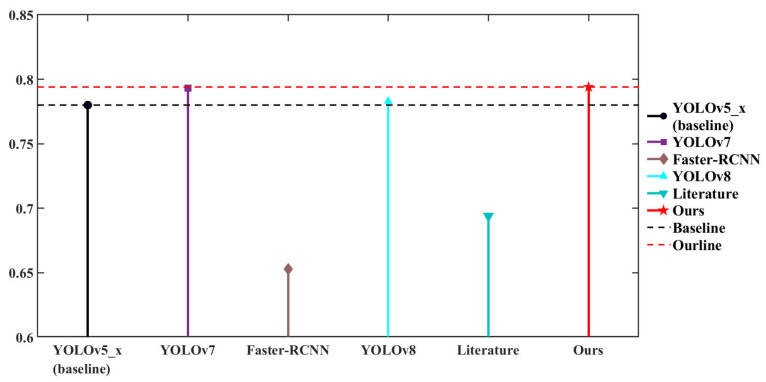
Visualization of comparative experiments based on URPC2020 dataset.

**Figure 20 sensors-23-03693-f020:**
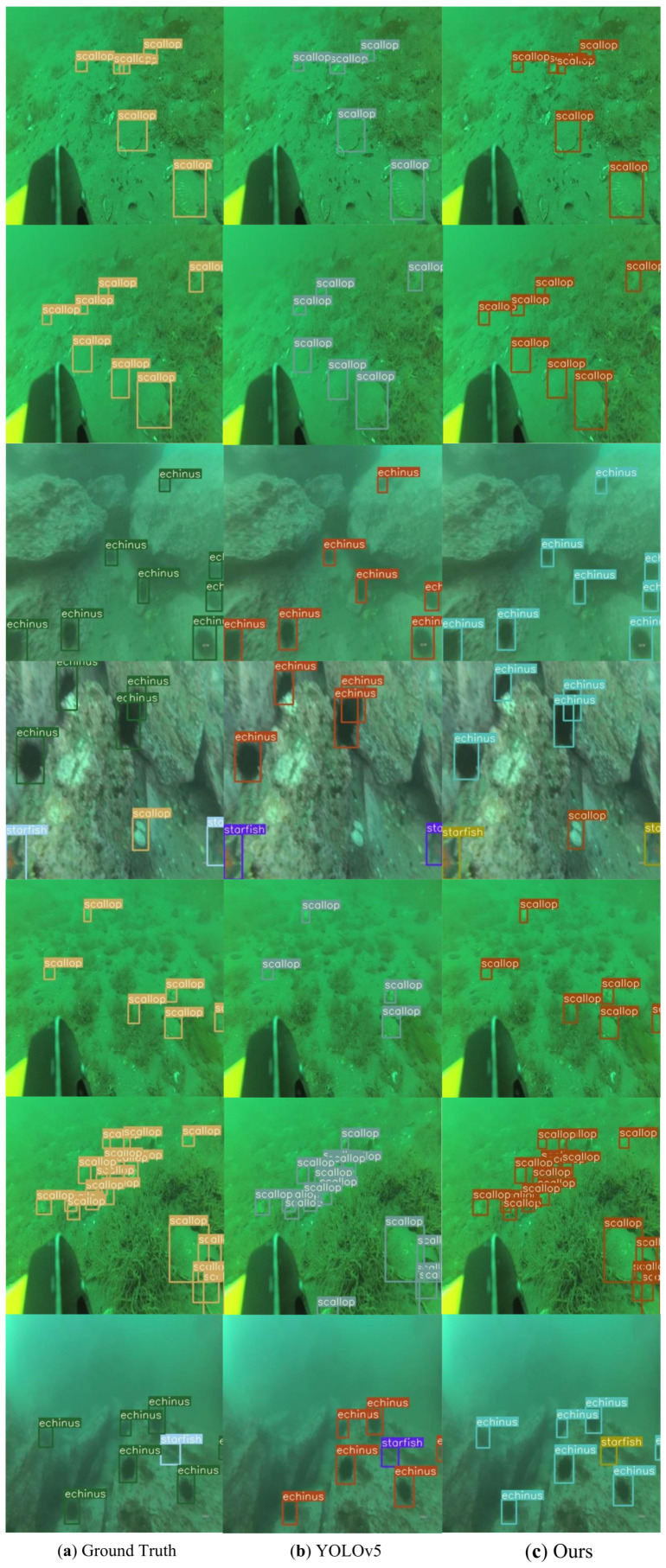
Underwater object-detection comparison results of YOLOv5 and improved YOLOv5.

**Table 1 sensors-23-03693-t001:** Ablation Study on URPC2019.

CSPLayer	BoT3	UDA	mAP@0.5	mAP@0.5:0.95
			0.753	0.413
✔			0.76	0.426
	✔		0.765	0.42
		✔	0.766	0.455
✔	✔		0.769	0.427
✔	✔	✔	0.798	0.442

**Table 2 sensors-23-03693-t002:** Comparative experiments based on the URPC2019 dataset.

Method	AP					mAP@0.5
	Echinus	Starfish	Holothurian	Scallop	Waterweeds	
YOLOv5_x (baseline)	0.924	0.885	0.731	0.826	0.4	0.753
YOLOv5_n	0.919	0.863	0.584	0.718	0.176	0.625
YOLOv5_m	0.921	0.886	0.729	0.822	0.34	0.74
YOLOv5_l	0.924	0.891	0.736	0.828	0.366	0.746
Faster-RCNN	0.8744	0.884	0.7896	0.701	0.6018	0.7708
YOLOv7	0.906	0.896	0.777	0.835	0.46	0.775
YOLOv8	-	-	-	-	-	0.76
**Ours**	**0.917**	**0.865**	**0.738**	**0.842**	**0.626**	**0.798**

**Table 3 sensors-23-03693-t003:** Comparative experiments based on the URPC2020 dataset.

Method	AP				mAP@0.5
	Holothurian	Echinus	Scallop	Starfish	
YOLOv5_x (baseline)	0.675	0.879	0.751	0.814	0.78
YOLOv7	-	-	-	-	0.793
Faster-RCNN	0.575	0.777	0.514	0.745	0.653
Literature [46]	-	-	-	-	0.694
YOLOv8	-	-	-	-	0.783
**Ours**	**0.694**	**0.879**	**0.777**	**0.825**	**0.794**

## Data Availability

The datasets generated during and/or analyzed during the current study are available from the corresponding authors upon reasonable request.

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
