# Peer review of "An Improved YOLOv5-Based Underwater Object-Detection Framework"

_sensors, 2023, doi:10.3390/s23073693_

Round 1
Reviewer 1 Report
Authors presented called “An Underwater Object Detection Framework Based on Improved YOLOv5" in their paper. In the study, the authors propose a marine biological object detection architecture based on an improved YOLOv5 framework. The authors claim that experiments on the URPC2019 and URPC2020 underwater datasets demonstrate that the proposed framework not only reduces the interference of underwater image degradation, but also offers superior performance for more sensitive detection in the original images in URPC2019 and URPC2020.My reviews and suggestions about their publications are listed ;
Authors should emphasize contribution and novelty in the abstract.
The introduction and related work sections should be extended to the more detailed background that would be supported by some literature. Add more recent reference to enhance literature survey section. Discuss the state-of-art techniques with their merits and issues. The literature should be developed and, if possible, presented in papers published in 2023.
Why did you do such a study? What is your contribution to the literature with this study? What is your difference from similar studies?
Some of your figure posts are too long. Please check throughout the article. For example “Figure 6. The CSPNeXt Block framework is divided into two different structures, A and B, according to whether add is True or False. The gray module is the DW convolution of the large kernel; however, in the improved framework of this paper, we use the ordinary convolution of the large kernel.”
It would be appropriate to include yolov8 in your work. The YOLOv8 model is designed to be fast, accurate, and easy to use, making it a good choice for a wide variety of object detection and image segmentation tasks, it can also be trained on large datasets and runs on a variety of hardware platforms from CPUs to GPUs.
The resolution of your figures should be increased.
The conclusion section is long. You'd better shorten it. Rewrite the conclusion with following comments:
- Highlight your analysis and reflect only the important points for the whole paper.
- Mention the implication in the last of this section. Please, carefully review the manuscript to resolve these issues.
Reviewer 2 Report
Dear Author
Manuscript entitled "An Underwater Object Detection Framework Based on Improved YOLOv5" Author have modified an existing pretrained model YoloV5. Author need to mention about challenges faced in existing YoloV5 for underwater object detection.
Mathematical model is not clear. need to explain with proper description.
Figure 12. Performance values for improved YOLOv5 model.... Graph/Plot should be mentioned with x axis and y axis with units.
Accuracy level for the proposed modified Yolov5 model of underwater object recognition should be mentioned in abstract and conclusion.
Reviewer 3 Report
The paper focuses on a novel idea with limitations. The following points need to be addressed in the revision:
- The title may be improved like, “An Improved YOLOv5-based Underwater Object Detection Framework”
- The abstract is in good shape. The terms without the proper name are not acceptable, like RTMDet, CSPLayer, etc. Rewrite the abstract after amendments.
- The result in the abstract, i.e. mAP@0.5 by 4.5%, needs to be posed in a more convincing way
- Comparison with SOTA techniques is not mentioned in the abstract
- The dataset, for results and worth of the algorithm determination, has not been cited and explained properly.
- The paper is lacking literature survey with recent methods. You should add references to the following recent articles in the introduction section:
https://doi.org/10.3390/app122010221
https://doi.org/10.3390/app121910036
https://doi.org/10.3390/app12199538
- Transformers section 2.3 needs more clarification concerning its architecture.
- In the data augmentation section, MLLE (Line 192) is a misfit. Make its provision, or move it to some appropriate position.
- Performance measures need to be detailed separately before results.
- Lots of typos and grammatical mistakes are in the article that needs to be corrected.
- Explain lines 396-404 concerning the accuracy measure fully explained.
- Table 2 contains unusual results that need to be explained.
- Table 2 and 3 have missing references while comparing your proposed method with SOTA techniques.
- The conclusion section needs revision with the concluding sentences of the article.
- Limitations and future work recommendations are missing and must be added as a separate section before Conclusions.
Reviewer 4 Report
An improved version of the YOLOv5 is presented to detect underwater objects. Results obtained for the URPC2019 and URPC2020 datasets show slight improvements in accuracy. In general, the work is exciting and well-written, however, some issues should be addressed to demonstrate novelties, contributions, and applicability.
Provide examples of images after the data augmentation module (line 183). MLLE method is only used in the data augmentation stage? or it should be always applied (real-time applications)? This is not clear.
Please highlight the differences between figure 1 (classic YOLOv5) and figure 3(improved YOLOv5).
Provide examples of figures in sections 4.1.1 and 4.1.2.
Justify the values shown in lines 347-352. Provide results for other values. Are they the best values?
For tables 1-3, add plots/graphs to observe maxima and minima values, trends, etc.
The computational cost has to be presented and discussed.
Provide the settings for the methods presented in tables 2 and 3. They have to be reproducible by readers. Also compare other qualitative and quantitative features.
In general, several new stages are added to the classic YOLOv5 network; however, the individual improvements of each stage are not clearly demonstrated. Only final results with slight improvements of 4.5 and 1.4% are presented. It is desired that authors show and demonstrate how the results improved after each newly added stage. For instance, which is the accuracy with and without the proposed data augmentation stage?
Can results for images different from those stored in the datasets be provided (your own photos)?
Analyze and discuss the limitations of your work.
Please use cursives for variables in the text.
Increase the font size in all the figures. This is fundamental for your work. Many labels in figures are unreadable.
Section 2.2 needs references.
Describe all the acronyms the first time they appear. In fact, please add a list of the abbreviations used.
Round 2
Reviewer 1 Report
I have reviewed the revised manuscript title " An Improved YOLOv5-based Underwater Object Detection Framework". After revising my initial comments and comparing the changes, done by the authors, with them. I found that the authors addressed and answered most of the comments efficiently. Overall, the revised manuscrip is well organized and carefully prepared. The response letter was elegant and satisfactory. I thank the authors for their kind responses. The authors have sufficiently address my all comments. So, I think it is appropriate to accept the revised article. The authors have addressed all the concerns and responded to the review comments. The manuscript can be published in this journal.
Reviewer 3 Report
Move Section 7 before Section 6.
The paper is in good shape now and may be accepted without modification.
Reviewer 4 Report
All comments and suggestions have been adequately addressed. This reviewer recommends the manuscript's acceptance.
